# Scene Change Detection with Vision-Language Representation Learning

## Abstract

Scene change detection (SCD) is crucial for urban monitoring and navigation, but remains challenging in real-world environments due to lighting variations, seasonal shifts, viewpoint differences, and complex urban layouts. Existing methods rely solely on low-level visual features, limiting their ability to accurately identify various changed objects amid the visual complexity of urban scenes. In this paper, we propose a vision-language framework for scene change detection that breaks through the single-modal bottleneck by incorporating semantic understanding through language. Our approach features a modular language component that leverages vision-language models (VLMs) to generate textual descriptions of detected changes, which are fused with visual features through a feature enhancer. Additionally, we introduce a geometric-semantic matching module that refines the predictions. To enable comprehensive evaluation, we present NYC-CD, a large-scale dataset of 8,122 real-world image pairs from New York City with multiclass change annotations, created through our semi-automatic annotation pipeline. Our method demonstrates strong performance across street-view benchmarks, achieving state-of-the-art results through semantic-visual feature integration. Extensive experiments demonstrate that our language module consistently improves existing change-detection architectures by substantial margins, highlighting the fundamental value of incorporating linguistic reasoning into visual change detection systems.

## 1 Introduction

Visual place recognition (VPR) seeks to identify a previously visited location from visual input. Although contemporary VPR systems perform well on short timescales, their precision deteriorates as the temporal gap between query and database images widens, as shown in Figure 1. Seasonal changes, construction, storefront updates, and vegetation growth gradually make the database less representative of the world, causing the retrieval recall to fall even when the place is the same. Our motivation is to confront this drift directly: *if we can detect what changed*, we can update or curate the database accordingly, and *rebound VPR performance toward its original level* without retraining the entire pipeline.

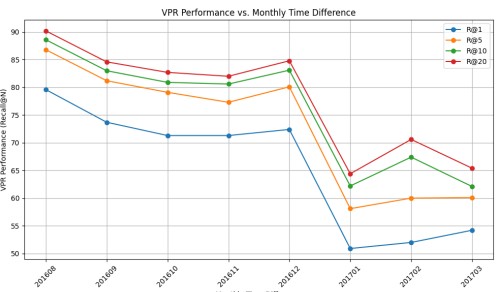

Figure 1: MixVPR (Ali-Bey et al., 2023) recall (R@1/5/10/20) vs. month offset evaluated on NYU-VPR dataset (Sheng et al., 2021). VPR performance degrades with time and drops sharply at the summer to winter transition, with persistent impact.

This perspective naturally foregrounds *scene change detection* (SCD): given two images of the same location captured at different times, determine what has changed and where alterations are evident in the captured frame. High–quality SCD provides the actionable signal needed for database maintenance and map updating: new objects can be incorporated, removed structures can be pruned, and appearance changes can be reflected in fresh exemplars. Crucially, the change signal should be *object–aware* (beyond coarse pixel differencing) so that updates target semantically meaningful entities rather than transient noise.

Table 1: Comparison of major scene change detection datasets with our NYC-CD.

| Datasets | Real/Sim | Indoor/Outdoor | Image Pairs | Target change | Non-target change |
|---|---|---|---|---|---|
| CD2014 (Wang et al., 2014) | Real | Outdoor | 7000 | dynamic object | – |
| PCD TSUNAMI (JST, 2015) | Real | Outdoor | 100 | structural change | weather, light |
| PCD GSV (JST, 2015) | Real | Outdoor | 100 | structural change | weather, light |
| VL-CMU-CD (Alcantarilla et al., 2018) | Real | Outdoor | 1362 | structural change | weather, light |
| PSCD (Sakurada et al., 2020) | Real | Outdoor | 770 | structural change | weather, light |
| CARLA-OBJCD* (Hamaguchi et al., 2020) | Sim | Outdoor | 15000 | new/missing object | light |
| GSV-OBJCD* (Hamaguchi et al., 2020) | Real | Outdoor | 500 | new/missing object | weather, light |
| Changesim (Park et al., 2021) | Sim | Indoor | ∼130,000 | new/missing/rotated/replaced | dusty air, low illumination |
| Standardsim (Mata et al., 2022) | Sim | Indoor | 12718 | new/missing object | – |
| UMAD (Li et al., 2024a) | Real | Outdoor | 26301 | anomalous object, dynamic object, new/missing object | weather, light |
| **NYC-CD (Ours)** | **Real** | **Outdoor** | **8122** | **new/missing object, appearance, new object due to viewpoint change** | **weather, light, anonymized people and car, motion blur** |

However, *urban* SCD is challenging. First, the change space is diverse—new/missing objects, construction progress, facade and billboard updates, foliage cycles, and dynamic traffic—often co-occurring within the same scene. Second, the acquisition is very often *unconstrained*: camera viewpoints differ between passes, along with parallax, scale changes, and occlusions that complicate correspondence. Third, illumination and weather introduce strong appearance shifts, and distractors such as reflections and shadows that mimic change. Finally, scalable supervision is difficult: large, real–world datasets with fine–grained object-level change labels are costly to obtain.

Existing solutions address subsets of the problem but leave important gaps. Simulation-based approaches (Park et al., 2021) provide controlled data, but struggle to transfer to the messiness of city streets. Fixed viewpoint methods (Wang et al., 2023b) bypass viewpoint variation by design, limiting their applicability to mobile platforms. Recent pipelines that employ vision foundation models (Kim & Kim, 2025) have improved robustness but remain largely uni-modal, relying on visual evidence alone and thus lacking priors, that help inform *what* has changed. As a result, predictions can be fragmented, sensitive to noise, or insensitive to subtle semantic changes.

In short, the gap is the lack of explicit, object-level priors about expected changes, priors that purely visual pipelines do not encode. Existing methods only consider uni-modal data, i.e., images, while neglecting to mine the rich semantic information available in multimodal data. This leaves the following question unclear: *Does applying a vision–language scheme enhance scene change detection?* Our hypothesis is that language-derived semantics—concise, structured descriptions of *what* changed—can act as priors that guide visual representations toward object-complete and contextually relevant masks. By combining *what* changed (language) with *where* changed (vision), the model should suppress distractors (e.g. reflections, shadows), disambiguate semantically subtle appearance shifts, and generalize across viewpoint and domain differences, ultimately producing cleaner predictions that better support long-term VPR-database updating.

The contributions of this paper are as follows:

- We propose a **vision-language framework for SCD** that injects language priors about changes and fuses them with visual features to produce precise object-aware change masks robust to illumination, appearance, and viewpoint shifts.

- We introduce a **scalable semiautomatic annotation pipeline** that enables a **large-scale and real-world dataset** with multi-class change labels, providing the diversity and scale necessary to study urban SCD at the object level.

- We present extensive experiments on street-view benchmarks, showing consistent gains over baselines and demonstrating that our language-informed SCD framework improves change-detection models to achieve **state-of-the-art performance on multiple datasets**.

## 2 RELATED WORKS

### 2.1 SCENE CHANGE DETECTION DATASETS

Table 1 highlights key differences between existing datasets and our proposed NYC-CD dataset. Most existing real-world datasets focus on limited change categories: CD2014 (Wang et al., 2014) targets only dynamic objects, while PCD TSUNAMI (JST, 2015), PCD GSV (JST, 2015), VL-

CMU-CD (Alcantarilla et al., 2018), and PSCD (Sakurada et al., 2020) concentrate on structural changes with pixel-level annotations. Object-level datasets like CARLA-OBJCD (Hamaguchi et al., 2020) and GSV-OBJCD (Hamaguchi et al., 2020) provide bounding box annotations for new/missing objects, while UMAD (Li et al., 2024a) expands to include anomalous and dynamic objects but still lacks appearance and viewpoint-induced changes. Recent simulated datasets—CARLA-OBJCD (Hamaguchi et al., 2020), Changesim (Park et al., 2021), and Standardsim (Mata et al., 2022)—offer large-scale data (up to 130,000 pairs) but lack real-world complexity, lighting variations, and unexpected occlusions. In contrast, our NYC-CD dataset provides 8,122 real-world outdoor image pairs with comprehensive change taxonomy covering new/missing objects, appearance changes, and viewpoint-induced changes, while offering segmented masks rather than bounding boxes for precise localization and excluding non-target changes such as weather, lighting, anonymized people, cars, and motion blur.

## 2.2 Scene Change Detection Methods

The majority of existing scene change detection methods share a fundamental limitation: reliance on uni-modal visual information. Supervised learning approaches, whether using convolutional neural networks (CNNs) (Varghese et al., 2018; Wang et al., 2023b; Park et al., 2022; Huang et al., 2023; Alcantarilla et al., 2018) or vision transformers (Lin et al., 2025; Wang et al., 2023a; Sachdeva & Zisserman, 2023; Huo et al., 2023), have achieved notable success in controlled settings. CNN-based methods such as ChangeNet (Varghese et al., 2018) and C-3PO (Wang et al., 2023b) establish strong baselines, while transformer-based methods such as RSCD (Lin et al., 2025) leverage self-attention mechanisms for improved long-range dependency modeling. However, both architectures remain constrained by relying on low-level visual features, which produce fragmented change masks that are disturbed by shadows and reflections while overlooking the semantic context of changed objects.

Recent advances in 3D Gaussian splatting (Lu et al., 2025; Jiang et al., 2025; Cho et al., 2025; Cheng et al., 2025; Zeng et al., 2025) and weakly/self-supervised learning (Sakurada et al., 2020; Li et al., 2024b; 2023; Lee & Kim, 2024; Hoyer et al., 2024; Ramkumar et al., 2021; Alpherts et al., 2025; Ramkumar et al., 2022) address specific challenges but maintain the uni-modal bottleneck. Gaussian splatting methods like 3DGS-CD (Lu et al., 2025) excel at handling geometric variations through 3D scene representations, yet require dense multiview captures that may not always be available. Weakly supervised approaches such as DiffMatch (Li et al., 2024b) and self-supervised methods such as EMPLACE (Alpherts et al., 2025) reduce annotation requirements but still learn exclusively from visual patterns. Training-free zero-shot methods (Kim & Kim, 2025; Kannan & Min, 2025) such as GeSCF offer impressive generalization using pre-trained models, but remain limited by visual-only features. Our approach overcomes this limitation by introducing a language module that integrates seamlessly with any existing architecture. By incorporating VLM-generated textual descriptions, adding our language and matching modules consistently boosts performance across all tested baselines with particularly dramatic improvements on complex urban scenes.

Although language-integrated change detection is emerging, existing methods differ significantly from our approach. ChangeCLIP (Dong et al., 2024) focuses on remote sensing applications with top-down perspective and relatively simple structured changes, such as building construction or deforestation, lacking the complexity of street-level variations. ViewDelta (Varghese et al., 2024) requires user-provided text prompts specifying expected changes, necessitating prior knowledge and potentially missing unexpected variations. Our method fundamentally differs by automatically generating comprehensive descriptions of all detected changes using GPT-4o, and then integrate text descriptions with visual features using our feature enhancer.

## 3 Annotation Method

Our annotation method aims to systematically identify and categorize scene changes between image pairs to create high-quality training data. Given a pair of images $I_0$ and $I_1$ captured at different times from potentially different viewpoints, our method annotates three distinct types of change on $I_1$ with respect to $I_0$: (1) new/missing objects, representing objects that appear or disappear between captures; (2) vegetation changes, capturing seasonal or growth-related alterations in plants and trees; and (3) objects not in view due to significant viewpoint change, distinguishing geometric

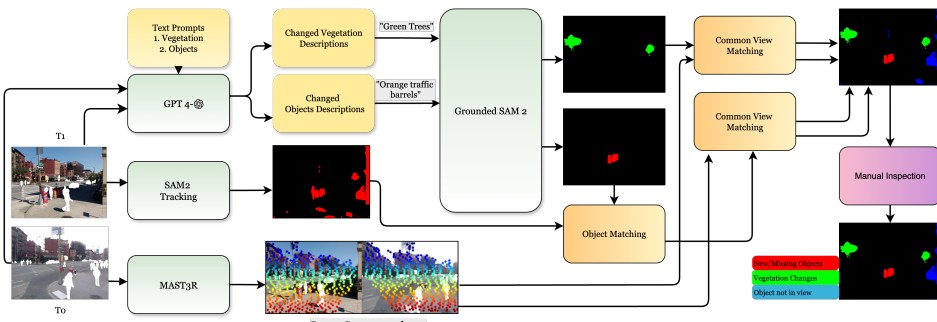

Figure 2: **Overview of our semi-automatic annotation pipeline.** GPT-4o generates descriptions of changed vegetation and objects from image pairs, which Grounded SAM 2 uses to segment changes. SAM2 Tracking identifies temporal inconsistent segments, and MAST3R estimates the common view between image pairs. Object Matching and Common View Matching filter noises and classify viewpoint-induced changes, producing initial masks. Manual inspection refines these masks into accurate ground truth for training and evaluation.

occlusions from actual scene changes. This multiclass annotation approach enables more nuanced understanding of urban scene dynamics compared to binary change detection.

Figure 2 presents the general pipeline of our annotation method. We dive into each module below.

**Change Captioning.** The annotation process begins by using GPT-4o to generate comprehensive descriptions of changes between pairs of images. We provide the model with $I_0$ and $I_1$ along with carefully crafted text prompts with one for vegetation and one for object changes (see Appendix A.1 for detailed prompts). In both cases, GPT-4o is explicitly instructed to exclude weather variations, lighting differences, and pedestrians (which are already anonymized in our dataset) to focus on semantically meaningful changes. The model is tasked with identifying objects and vegetation that appear in $I_1$ but not in $I_0$. For vegetation changes, we specifically request descriptive attributes such as "green trees" or "bare bushes" to capture seasonal variations. This process yields two distinct lists: changed objects with their descriptions and changed vegetation with appearance attributes.

**Open Vocabulary Segmentation.** The descriptive lists from GPT-4o are processed through Grounded SAM 2 (Ren et al., 2024), which combines Grounding DINO's open-set object detection capabilities with SAM's precise segmentation. This architecture enables detection and segmentation of arbitrary regions based on text input, making it ideal for our diverse urban scenes. Grounded SAM 2 segments all instances of objects from the GPT-generated list in $I_1$, regardless of whether they represent actual changes. For instance, if GPT identifies "chairs" as a change due to additional chairs appearing in $I_1$, Grounded SAM 2 segments both preexisting and new chairs. Similarly, vegetation is segmented using descriptive phrases like "bare trees" or "green foliage," where the descriptive nature inherently indicates that these segments represent changes.

**SAM2 Tracking.** To identify temporally inconsistent regions, we use SAM2 (Ravi et al., 2024) tracking capabilities across the image pair. The process begins with automatic segment generation in $I_1$, producing comprehensive masks of objects. These segments are then used as prompts for $I_0$, allowing us to identify segments present in $I_1$ but absent in $I_0$—potential indicators of scene changes. However, this raw output often includes fractured or incomplete object masks and irrelevant changes such as shadows or reflections. These temporal inconsistency masks require refinement through matching with the semantically meaningful masks from Grounded SAM 2.

**Object Matching.** We refine the change detection by computing the overlap between SAM2 tracking masks and Grounded SAM 2 object masks. For each SAM2 mask, we calculate the intersection ratio $\alpha$ with Grounded SAM masks, retaining only those where $\alpha > \alpha_t$ (a predefined threshold). This matching process leverages the complementary strengths of both approaches: SAM2 tracking identifies changing regions but may include irrelevant variations, while Grounded SAM 2 identifies

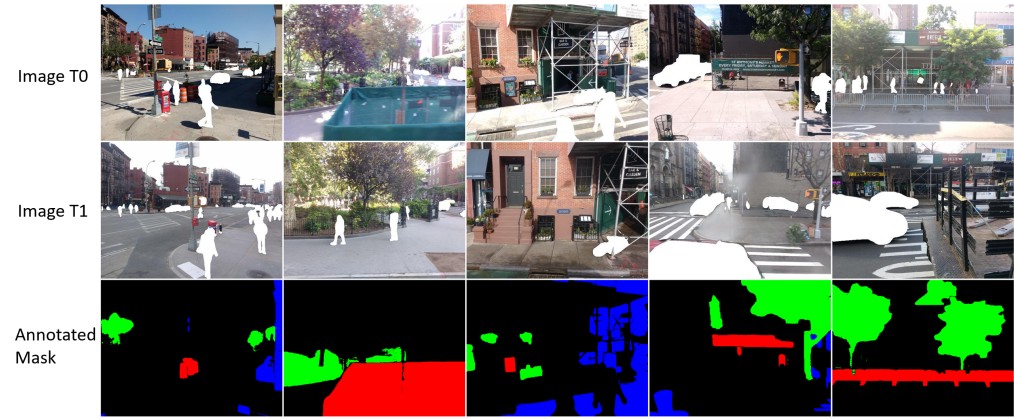

Figure 3: 5 image pairs (column) with multi-class change detection mask annotated by our pipeline.

semantically relevant objects that are potentially changing regions. The intersection yields high-confidence masks that represent the actual changes in relevant objects.

**Common View Estimation and Matching.** To distinguish between actual scene changes and viewpoint-induced variations, we employ MAST3R (Leroy et al., 2024) for robust dense matching between image pairs. MAST3R excels in two-view regimes, predicting dense correspondences and estimating the common-view region that is the area visible in both images. We calculate the overlap between each filtered object mask and the common-view mask. Objects overlapping with the common view are classified as genuine new/missing object changes, while those outside are categorized as objects not in view. The same classification process applies to the vegetation masks in Grounded SAM 2, distinguishing actual vegetation changes from viewpoint-induced occlusions. This step produces our initial pseudo-masks with multi-class labels.

**Manual Inspection.** The final stage involves careful human verification of the initial pseudo-masks to ensure annotation quality. We systematically review each pair of images, discarding those with false negatives (unlabeled changes in the initial masks) to maintain the completeness of the data set. False positive masks are manually removed to ensure precision. This semiautomatic approach efficiently combines the scalability of automated methods with the accuracy of human judgment, producing high-quality multiclass change detection annotations for complex urban scenes. Figure 3 shows the multiclass change mask annotated by our semiautomatic method for different image pairs.

## 4 THE NYC-CD DATASET

The NYC-CD dataset comprises 8,122 carefully curated image pairs extracted from the NYU-VPR dataset (Sheng et al., 2021), providing a comprehensive benchmark for urban scene change detection. The source NYU-VPR dataset contains street-level imagery recorded throughout Manhattan, New York from May 2016 to March 2017, captured using smartphone cameras mounted on the front, back and side of fleet vehicles. All images include GPS tags for spatial reference, while pedestrians and vehicles are replaced with white pixels using MSeg (Lambert et al., 2020) to preserve privacy. They are . This rich temporal and spatial coverage of one of the world's most dynamic urban environments provides an ideal foundation for studying complex scene changes.

To construct meaningful image pairs for change detection, we employ visual place recognition (VPR) instead of simple GPS matching. VPR retrieves the most visually similar images for a query from a database of images with known camera poses, ensuring sufficient visual overlap between the paired images. We specifically use images from different quarters as database and queries (Q1 paired with Q3, Q2 paired with Q4), guaranteeing a minimum three-month temporal gap between image pairs to capture meaningful scene changes. Using MixVPR (Ali-Bey et al., 2023), a state-of-the-art VPR method, we pair each database image with its top-ranked retrieval result to form potential change pairs. This VPR-based approach is crucial because GPS coordinates alone would often yield image pairs with minimal or no viewpoint overlap, making change detection infeasi-

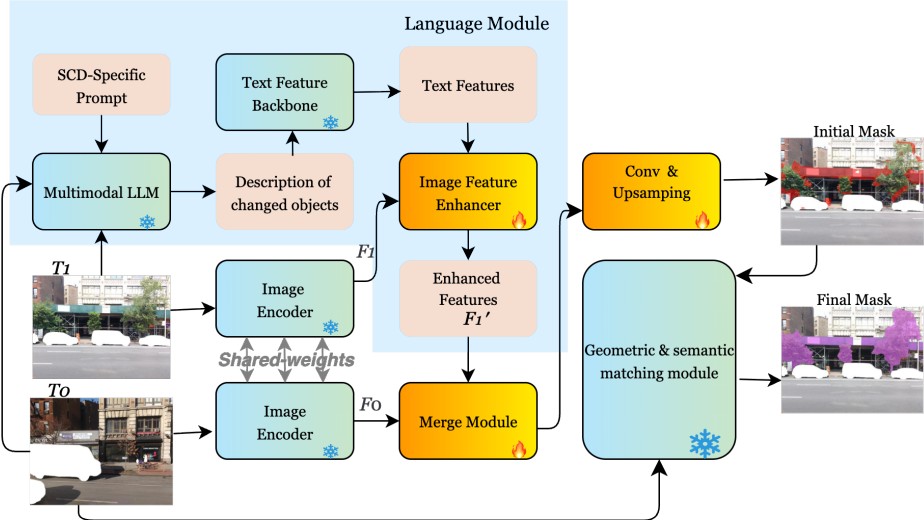

Figure 4: **Overview of the proposed architecture.** Image 0 and Image 1 are processed by a shared-weight image encoder to extract features $F_0$ and $F_1$. Multimodal LLM/GPT-4o generates descriptions of changed objects in Image 1 relative to Image 0. A text feature backbone encodes these descriptions into text features, which are used by an image feature enhancer to augment $F_1$, producing enhanced features $F_1'$. The enhanced features are merged with $F_0$ and passed through a segmentation head to produce the initial change mask. SAM's class-agnostic masks and Grounding SAM masks are then leveraged to refine the initial mask using both geometric and semantic cues.

ble. By prioritizing visual similarity while maintaining temporal separation, we ensure that each potential pair contains sufficient common content for meaningful change analysis while capturing the natural viewpoint variations that occur in real-world scenarios.

The NYC-CD dataset introduces several unique characteristics that distinguish it from existing benchmarks. Most notably, it explicitly incorporates and annotates viewpoint-induced changes, addressing a critical gap in current datasets that either assume static cameras or attempt to minimize viewpoint differences. In practical applications like autonomous navigation and augmented reality, viewpoint variations are inevitable and can create apparent changes that are purely geometric rather than semantic. By categorizing these as a distinct change type, our dataset enables algorithms to differentiate genuine scene changes from perspective-induced variations. Furthermore, the dataset captures the full complexity of dense urban environments with large temporal spans. The urban environment is rich with diverse changes, including seasonal foliage variations, ongoing construction, evolving storefront appearances, and dynamic objects in motion. Unlike controlled simulated environments that cannot replicate real-world unpredictability, NYC-CD provides authentic urban scene dynamics, making it an essential resource for developing robust change detection systems capable of handling the complexities of real-world deployment.

## 5 METHODOLOGY

### 5.1 OVERALL ARCHITECTURE

We introduce an innovative methodology that integrates vision-language models into scene change detection, addressing the fundamental limitations of uni-modal approaches. As illustrated in Figure 4, our architecture consists of two key components: a language module and a geometric-semantic matching module that can be seamlessly integrated into any existing change detection model to enhance its performance. The base architecture follows a standard supervised change detection pipeline: a shared weight image feature backbone extracts features $F_0$ and $F_1$ from the input image pair $(I_0, I_1)$, which are then processed through cross-attention mechanisms and merged to form a unified feature representation. This merged feature passes through a segmentation head to produce

the initial change mask. Our contribution lies in two critical enhancements: (1) the language module that enriches the image feature $F_1$ with semantically aware text features before merging with $F_0$, effectively compensating for the inherent limitations of visual-only representations, and (2) the geometric-semantic matching module that refines the initial predictions by leveraging both SAM's class-agnostic object proposals and Grounded SAM's semantically informed segmentation, ensuring detected changes are both complete and contextually accurate.

## 5.2 LANGUAGE MODULE

The language module leverages the powerful visual-linguistic understanding capabilities of GPT-4o to inject semantic awareness into the change detection process. Given an image pair, GPT-4o first generates comprehensive descriptions of objects that appear in $I_1$ but not in $I_0$, effectively identifying potential changes through natural language. These textual descriptions are then processed by a text feature backbone to extract rich semantic features. The image feature enhancer takes the original image features $F_1$ and enhances them with the extracted text features to produce semantically enriched features $F_1$'.

The feature enhancer architecture, adapted from GroundingDINO (Liu et al., 2024), consists of multiple stacked enhancement layers that facilitate cross-modal fusion. Each layer employs deformable self-attention for processing image features, allowing the model to adaptively focus on relevant spatial regions, while vanilla self-attention is used for text features to maintain semantic coherence. We incorporate bidirectional cross-attention mechanisms, image-to-text and text-to-image, that enable deep feature alignment between modalities. The image-to-text attention allows visual features to query relevant semantic information, while text-to-image attention grounds textual concepts in visual regions. This bidirectional information flow ensures that the enhanced features $F_0$' contain both precise spatial localization from the visual domain and rich semantic understanding from the language domain, significantly improving the model's ability to distinguish meaningful changes from irrelevant variations.

## 5.3 GEOMETRIC AND SEMANTIC MATCHING MODULE

The geometric and semantic matching module elevates change detection from pixel-level analysis to object-level understanding by refining the initial segmentation masks through complementary matching strategies. This module addresses two critical challenges in change detection: incomplete object segmentation and noise from irrelevant changes.

The geometric matching component leverages SAM2's tracking capabilities to ensure spatial completeness of detected changes (Ravi et al., 2024). By generating class-agnostic object proposals and tracking them across the image pair, SAM2 identifies regions with temporal inconsistency. We evaluated the overlap ratio $\alpha$ between each SAM tracking mask and the initial prediction, retaining only those masks where $\alpha > \alpha_t$ (a predefined threshold). This geometric matching effectively completes fragmented object segments that may be partially detected in the initial mask, ensuring that entire objects are identified as changed rather than just portions of them.

The semantic matching component employs Grounded SAM (Ren et al., 2024) to filter out noise and validate semantic relevance. Using the change descriptions generated by GPT-4o, Grounded SAM produces semantically informed segmentation masks for the identified objects. By computing overlap ratios between these semantic masks and the initial predictions, we can distinguish between genuine object-level changes and spurious detections caused by shadows, reflections, or other irrelevant variations. Masks that satisfy both geometric and semantic matching criteria represent high-confidence, semantically meaningful changes (see Appendix Figure 8 for qualitative results).

The synergy between these two matching strategies is crucial: geometric matching ensures spatial completeness, while semantic matching guarantees relevance. This dual validation process transforms noisy, incomplete pixel-level predictions into clean, comprehensive object-level change masks, significantly improving both the precision and interpretability of the detection results. The final output provides not just binary change information but semantically rich, object-aware change

Table 2: F1-score and IoU comparison of GeSCF, RSCD, and C-3PO models with and without the language module on three datasets: Ours, VL-CMU-CD, and PSCD. Bold numbers indicate highest performance per column. Delta values in parentheses show the effect of adding the language module.

| Model | Language | Our Dataset | | VL-CMU-CD | | PSCD | |
|-------|----------|-------------|-----|-----------|-----|------|-----|
| | | F1 | IoU | F1 | IoU | F1 | IoU |
| GeSCF | No | 0.16 | 0.10 | 0.77 | 0.65 | 0.40 | 0.28 |
| GeSCF | Yes | 0.57 (+0.41) | 0.49 (+0.38) | 0.82 (+0.05) | 0.71 (+0.06) | 0.49 (+0.09) | 0.37 (+0.09) |
| RSCD | No | 0.13 | 0.07 | 0.83 | 0.73 | 0.54 | 0.40 |
| RSCD | Yes | **0.72** (+0.59) | **0.62** (+0.55) | **0.85** (+0.02) | **0.75** (+0.02) | 0.59 (+0.05) | 0.46 (+0.06) |
| C-3PO | No | 0.09 | 0.10 | 0.50 | 0.39 | **0.67** | **0.55** |
| C-3PO | Yes | 0.57 (+0.48) | 0.47 (+0.37) | 0.73 (+0.23) | 0.67 (+0.28) | 0.57 (-0.10) | 0.44 (-0.11) |

detection that better serves downstream applications in urban monitoring and autonomous navigation.

## 6 RESULTS

### 6.1 SETUP

We evaluated our proposed method against three representative baselines that span different architectural paradigms in change detection. C-3PO (Wang et al., 2023b) represents traditional CNN-based approaches, employing pixel-level change detection through convolutional neural networks. RSCD (Lin et al., 2025) exemplifies modern transformer-based methods, utilizing DINOv2 (Oquab et al., 2023) features with full image cross-attention mechanisms. GeSCF (Kim & Kim, 2025) demonstrates zero-shot capabilities by using pre-trained SAM without task-specific training.

Our evaluation encompasses seven diverse datasets that collectively challenge change detection methods in diverse scenarios and perspectives. For street view evaluation, we used our NYC-CD dataset, VL-CMU-CD, PSCD, and ChangeSim. These datasets were specifically chosen for their scene diversity, ranging from natural disasters to gradual urban development and from real-world to simulated environments. For fair comparison, we carefully filter all datasets to exclude biased ground truth annotations (see Appendix Figure 6).

### 6.2 RESULTS

**Quantitative comparison.** Table 2 presents comprehensive quantitative comparisons across three diverse datasets, demonstrating the substantial impact of our language-enhanced approach. In our challenging NYC-CD dataset, the integration of language and matching modules yields dramatic improvements across all baseline methods. RSCD achieves the highest performance with our enhancements, yielding nearly a five-fold increase in detection accuracy. Similarly, impressive gains are observed for GeSCF and C-3PO, highlighting that our modules effectively address the complex urban changes present in the dataset. In VL-CMU-CD, our approach achieves state-of-the-art results, with RSCD + language reaching 0.85 F1 and 0.75 IoU. The consistent improvements across all baselines on this dataset demonstrate our method's robustness to different architectural backends. Although PSCD shows more modest gains, this likely reflects the dataset's focus on disaster-induced changes, where geometric cues may be more critical than semantic understanding. In general, these results validate that the incorporation of language-based semantic reasoning substantially enhances change detection performance, particularly for complex urban environments with diverse changes.

Multi-class change detection analysis in Table 3 corroborates our approach. The RSCD baseline shows varying performance across categories, with dynamic changes proving more challenging. Integrating our modules consistently improves results across all classes, with particularly strong gains in the most difficult categories such as object not in view (see Appendix Figure 9). Overall, these enhancements yield higher mean performance and demonstrate more balanced and robust detection, especially in scenarios involving geometric transformations and appearance variations.

**Qualitative comparison.** In Figure 5, our method demonstrates consistently cleaner, object-

| Class / Summary | F1-score | | IoU | |
|---|---|---|---|---|
| | RSCD | RSCD + Our Modules (Δ) | RSCD | RSCD + Our Modules (Δ) |
| 0. Non-change area | 0.899 | 0.945 (+0.046) | 0.816 | 0.895 (+0.079) |
| 1. New/Missing Object | 0.438 | 0.472 (+0.034) | 0.281 | 0.309 (+0.028) |
| 2. Vegetation Change | 0.655 | 0.703 (+0.048) | 0.487 | 0.542 (+0.055) |
| 3. Object not in view | 0.459 | 0.563 (+0.104) | 0.298 | 0.391 (+0.093) |
| **Mean (Macro F1 / mIoU)** | 0.613 | 0.671 (+0.058) | 0.471 | 0.534 (+0.063) |

Table 3: Comparison of RSCD and RSCD with our modules on multi-class change detection.

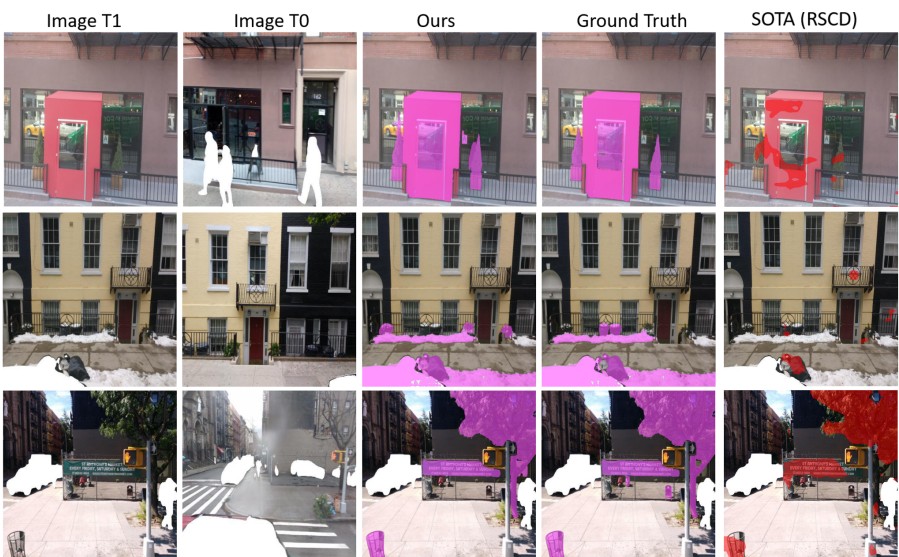

Figure 5: **Comparative results of the current state-of-the-art model and our method under various environments.** Our method outperforms with precise boundaries and edges, where the state-of-the-art model hardly captures changes.

complete predictions than the current SOTA (RSCD unified model) in varied urban scenes. Whereas the baseline often misses subtle changes or produces fragmented blobs, our results exhibit crisp boundaries and accurate edges that align with object extents. The predictions remain robust to misleading cues—such as glass reflections, shadows, and transient clutter—allowing the model to localize multiple changes within the same frame without crosstalk. Crucially, the approach captures diverse change types beyond new/missing objects, including vegetation and broader appearance alterations, illustrating stronger robustness under illumination and viewpoint variations.

# 7 CONCLUSION

In this work we tackle urban scene change detection under long-term, cross-view, and seasonal variation by coupling vision backbones with language-derived semantics. We contribute (i) a vision–language framework that fuses text cues about what has changed with image features to guide fine-grained segmentation, (ii) a scalable semi-automatic annotation pipeline that enables a large real-world dataset with multiclass change labels, and (iii) an extensive evaluation showing consistent gains in street-view benchmarks. Together, these results demonstrate that injecting language priors overcomes the limits of purely visual methods, improving robustness to illumination, appearance, and viewpoint shifts while providing a practical path to scalable supervision. Looking ahead, we will reduce residual annotation bias, strengthen robustness to localization/viewpoint noise, and move toward an open, foundation-style model that transforms both street-view and remote-sensing change detection.

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

## A APPENDIX

### A.1 TEXT PROMPTS TO GPT-4O

**Text prompt for object changes:** I have 2 images of outdoor scenes: A and B. In this task, you must not talk about weather, lighting, vehicles, people, or animals. Refrain from mentioning any elements that are not directly observable or are obscure. You must try your best to find all objects you see in A. For every object you see in A, you should try your best to find a corresponding object in B. Then, you must try your best to find all objects you see in B. For every object you see in B, you should try your best to find a corresponding object in A. In your response, just give me a numbered list of objects that you fail to find a match in B, and a numbered list of objects that you fail to find a match in A. Do not use prepositions.

**Text prompt for vegetation changes:** I have 2 images of outdoor scenes: A and B. In this task, you must try your best to find all vegetations you see in A. For every plant you see in A, you should try your best to find if it changed in B. Then, you must try your best to find all vegetations you see in B. For every plant you see in B, you should try your best to find if it changed in A. In your response, just give me a numbered list of changed plant names in A, and a numbered list of changed plant names in B. If the plant is removed, do not include it in the list. If there are no visible changed plants in the image A, write "1. None". If there are no visible changed plants in the image B, write "1. None". If a plant is changed, only include the plant name after change. Do not use prepositions such as to. For example: green trees, bare trees, green bushes, bare bushes, potted plants, etc.

## A.2    NOISY LABEL EXAMPLES FROM VL-CMU-CD DATASET

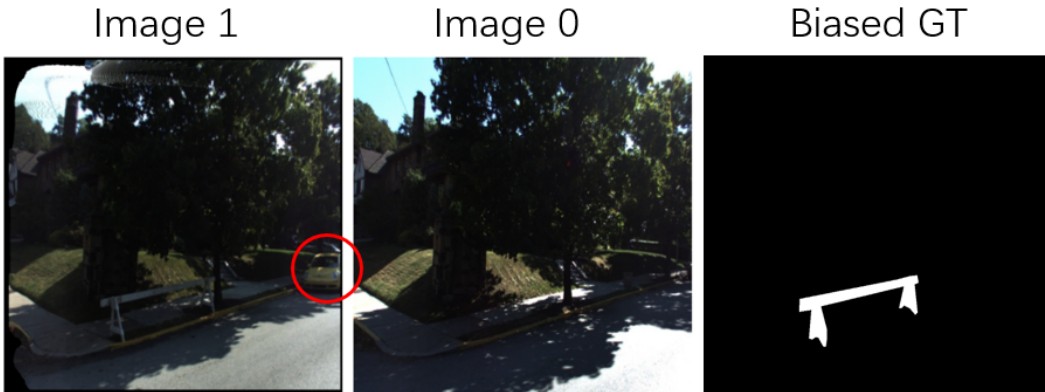

Figure 6: Example of noisy label. The image pairs are from VL-CMU-CD and the changed car is not annotated.

## A.3    EVALUATION METRIC

We employ two standard metrics for the quantitative evaluation of binary change detection performance: Intersection over Union (IoU) and F-1 score. IoU measures the overlap between predicted and ground-truth masks by dividing the intersection (pixels correctly identified as changed) by the union (all pixels marked as changed in either prediction or ground-truth). The F-1 score, calculated as the harmonic mean of precision and recall, provides a balanced measure of detection precision. Higher F-1 score and IoU indicate better alignment with ground truth (Figure 7).

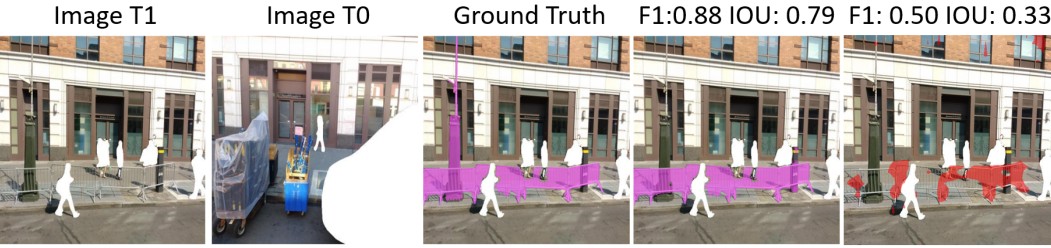

Figure 7: Higher IOU or F-1 score indicate better alignment with ground truth.

## A.4 QUALITATIVE RESULTS FOR MATCHING MODULE

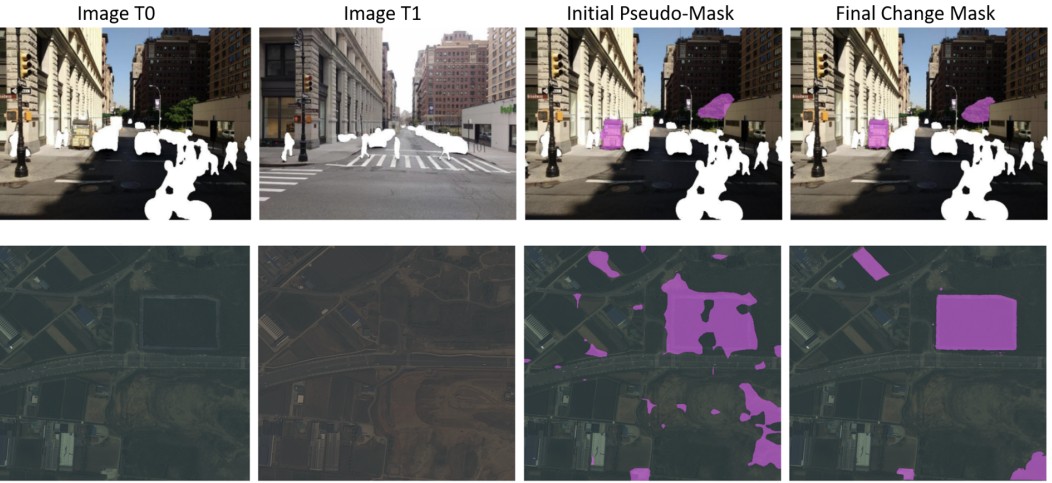

Figure 8: Qualitative results of the matching module. Each column shows Image $t_0$, Image $t_1$, the initial pseudo-mask, and the final change mask. The module suppresses spurious responses and yields complete, object-accurate masks.

## A.5 QUALITATIVE RESULTS FOR MULTI-CLASS CHANGE DETECTION

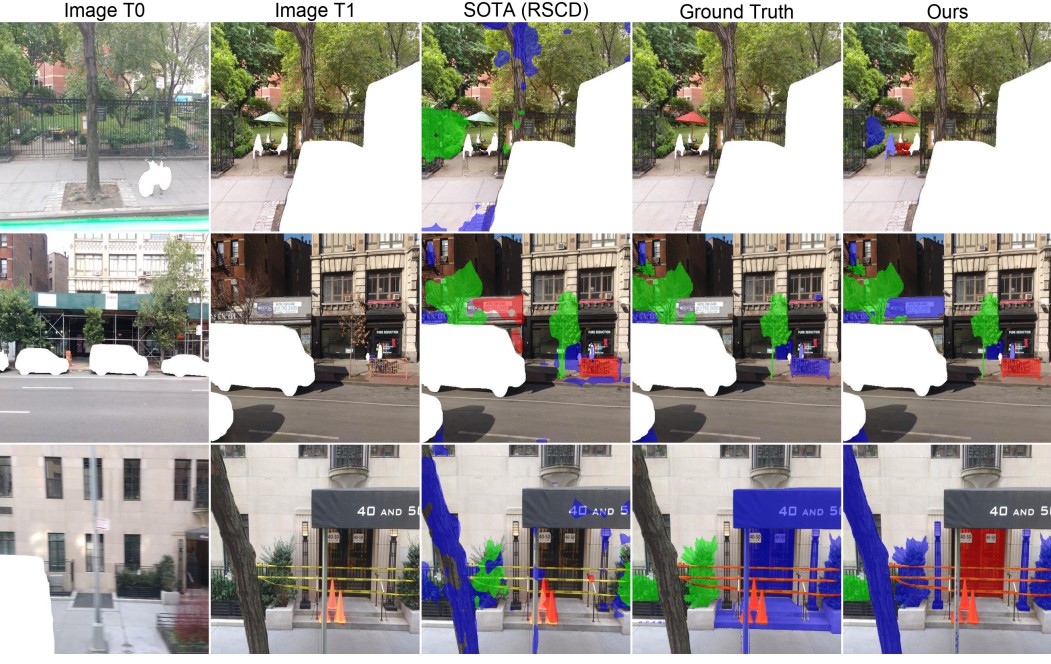

Figure 9: Additional comparative results of our multi-class change detection method with ground truth and current SOTA method shows our method also outperforms significantly in complex urban environment as well.

## A.6 CROSS-DOMAIN GENERALIZATION RESULTS

Figure 10: Improvement of the language module on F1-score and IoU across multiple domains. Results are from a *unified model* trained jointly on seven datasets—four street-view (NYC-CD, VL-CMU-CD, PSCD, ChangeSim) and three remote sensing (S2Looking, SYSU-CD, CDD).

We include three remote sensing datasets: S2Looking (Shen et al., 2021), SYSU-CD (Shi et al., 2021), and CDD (Lebedev et al., 2018) to assess cross-domain generalization. The generalization capability of our language module is further validated through multi-domain experiments combining street view and remote sensing datasets (Figure 10). Training a unified model across all seven datasets: four street view datasets (NYC-CD, VL-CMU-CD, PSCD, ChangeSim) and three remote sensing datasets (S2Looking, SYSU-CD, CDD) reveal consistent performance improvements regardless of imaging perspective or domain. The language module improves performance by approximately 10% on average in all datasets, with particularly strong gains in our NYC-CD dataset (over 10% improvement in both F1 and IoU) and VL-CMU-CD (approximately 12% improvement in F1). Even remote sensing datasets, which have fundamentally different viewing angles and change characteristics compared to street-level imagery, benefit from our semantic enhancement, with improvements ranging from 5% to 15% in F1 scores. This cross-domain effectiveness demonstrates that the semantic understanding provided by language transcends specific imaging modalities, offering a domain-agnostic solution to improve change detection. The consistent improvements across such diverse datasets, from disaster assessment to urban monitoring, from ground-level to aerial perspectives, underscore the fundamental value of incorporating linguistic reasoning into visual change detection, suggesting that semantic context is universally beneficial regardless of the specific application domain or imaging conditions.

