# OpenReview forum: "Scene Change Detection with Vision-Language Representation Learning"
_ICLR.cc/2026/Conference — ICLR 2026 Conference Withdrawn Submission_

### Official Review · Reviewer_EqbL · 2025-10-22

**Soundness:** 3
**Presentation:** 2
**Contribution:** 3
**Rating:** 4
**Confidence:** 3

**Summary:**

- (1) The paper addresses critical challenges in Scene Change Detection (SCD), including 1) performance degradation caused by lighting variations, seasonal shifts, viewpoint differences, and etc., 2) existing methods’ over-reliance on unimodal low-level visual features.
- (2) To solve above issue, the authors propose a vision-language framework consisting of two core modules: (1) a language module and (2) a geometric-semantic matching module.
- (3) Additionally, the authors construct the NYC-CD dataset (8,122 real-world New York City street-view image pairs) with multiclass change annotations via a semi-automatic pipeline.
- (4) Key validation methods include extensive experiments on 7 datasets (4 street-view, 3 remote sensing) comparing the proposed framework with 3 baseline models (CNN-based C-3PO, Transformer-based RSCD, zero-shot GeSCF) using F1-score and IoU metrics.

**Strengths:**

> 1. The proposed framework addresses the unimodal bottleneck of existing SCD methods by introducing a “plug-and-play” language module and geometric-semantic matching module. The framework automatically generates change descriptions via GPT-4o and fuses multimodal features, enabling end-to-end adaptation to complex street-view scenes. Experimental results confirm its generality.
> 2. NYC-CD provides 8,122 real outdoor image pairs with multiclass annotations. NYC-CD better reflects real urban dynamics, making it a valuable benchmark for object-level SCD research.
> 3. Technical details are sufficiently detailed: e.g., the semi-automatic annotation pipeline in Section 3 breaks down GPT-4o captioning, Grounded SAM segmentation, and MAST3R common-view estimation step-by-step. This clarity facilitates reproducibility of the work, easy to understand.

**Weaknesses:**

>1. The paper relies heavily on GPT-4o for generating change descriptions, but lacks critical analysis of GPT-4o’s impact on annotation and framework performance. For example, it does not verify how GPT-4o’s description accuracy (e.g., missed or false change descriptions) affects subsequent Grounded SAM segmentation or final SCD results.
> 2. While the paper validates the overall framework’s effectiveness, it lacks ablation experiments to isolate the contributions of individual components.
>>For instance, it does not verify:
>>>- (1) the independent impact of the language module (without geometric-semantic matching) or vice versa
>>>- (2) the necessity of bidirectional cross-attention in the feature enhancer (Section 5.2), or
>>>- (3) the sensitivity of the overlap threshold αₜ (Section 5.3) to final performance. This makes it difficult to determine which components drive the performance gains. Ablation experiments are not sufficient.

**Questions:**

>- 1. How would the framework perform on image pairs with extreme viewpoint differences (e.g., <30% overlap, common in real-world mobile navigation)? Could the authors supplement experiments on a subset of NYC-CD with controlled low-overlap pairs to verify the framework’s viewpoint robustness boundary?
>- 2. In the geometric-semantic matching module, SAM2 tracking is used to identify temporally inconsistent regions, but the paper notes this may produce “fractured or incomplete masks” . Did the authors test alternative temporal consistency methods and compare their performance with SAM2? If not, why is SAM2 the optimal choice for this task, especially considering its potential computational cost in large-scale SCD?
>- 3. The NYC-CD dataset is constructed from the NYU-VPR dataset (2016–2017), which has a relatively short temporal span (11 months). Would the framework’s performance degrade on image pairs with longer temporal gaps? Could the authors discuss the framework’s scalability to long-term SCD or supplement experiments on a dataset with longer time intervals?
>- 4. The language module uses GPT-4o, a proprietary and computationally expensive large language model (LLM). For practical deployment (e.g., real-time urban monitoring on edge devices), how would the authors optimize the framework’s inference speed? Have they tested lighter-weight open-source LLMs (e.g., LLaVA-1.5, Vicuna) as alternatives to GPT-4o, and if so, how do these alternatives affect SCD performance and efficiency?
>- 5. Several key symbols and parameters lack clear definitions, hindering reproducibility. For example: (1) In Section 5.3, the “predefined threshold αₜ” for geometric matching is not specified (e.g., whether αₜ = 0.5, and how to tune?); (2) In Section 4, the “VPR-based image pairing” process mentions using MixVPR but does not clarify the retrieval top-k setting (e.g., it is top-1?) or how to filter low-overlap pairs (if have this issue)?

---

### Official Review · Reviewer_h58H · 2025-10-27

**Soundness:** 3
**Presentation:** 3
**Contribution:** 3
**Rating:** 6
**Confidence:** 3

**Summary:**

The a proposes a vision-language framework for Scene Change Detection (SCD) that augments a standard two-branch SCD backbone with (i) a language module that uses a VLM (GPT-4o) to auto-describe changes and fuses text features with image features via a feature-enhancer, and (ii) a geometric–semantic matching module that refines masks using SAM2 tracking and Grounded-SAM (open-vocabulary segmentation). The authors also introduce NYC-CD, a real-world street-view dataset of 8,122 image pairs with multi-class labels (new/missing objects, vegetation change, object not in view). Across several benchmarks, language-guided models show notable gains over visual-only baselines (e.g., RSCD, C-3PO, GeSCF).

**Strengths:**

- The proposed language-guided and geometric modules drop cleanly into standard SCD backbones with minimal surgery. In practice, they behave like wrap-around add-ons that deliver steady, repeatable gains across datasets rather than one-off wins tuned to a single model.

- The NYC-CD corpus (8,122 street-view pairs) includes a crucial not-in-view label, separating true appearance/disappearance from viewpoint changes. A semi-automatic pipeline captioning, text-conditioned proposals, geometric checks followed by human verification strikes a practical balance between scale and annotation quality.

- Geometric semantic refinement with SAM2 + Grounded-SAM is practical and improves object-completeness vs. blob-like masks.

- Consistent gains on RSCD, C-3PO, and GeSCF indicate the approach is not tied to a particular architecture. The language prior complements both foundation-feature and segmentation-style pipelines, suggesting it should transfer well to future SCD backbones, too.

**Weaknesses:**

- The proposed pipeline significantly depends on GPT-4o both for dataset labels and for online captions. Please specify exact prompts (Appendix A.1 helps), temperature/seeds, and whether an open alternative (e.g., Llava-Next, InternVL) yields similar gains.

- The cost/latency of generating captions + text encodings at train/test time and of running SAM2/Grounded-SAM/MASt3R should be reported; ablate the feature-enhancer depth and text-encoder size.

- In ablations author should provide: (i) without text fusion (visual-only), (ii) with user-prompted text (ViewDelta-style) vs auto captions, (iii) with/without each matching submodule (SAM2 vs Grounded-SAM vs both), (iv) α-threshold sweep.

- In the comparisons authors need to add direct comparisons to ViewDelta (text-prompted) and ChangeCLIP (VLM-guided CD) to clarify advantages of auto-captioning + fusion and of the street-view focus.

- Since GeSCF emphasizes cross-domain robustness and also releases ChangeVPR, evaluate or discuss your method there to position your contributions more sharply.

**Questions:**

- State the exact text encoder (e.g., CLIP ViT-B/16, SigLIP-text) and how tokens map to image space (per-token cross-attention vs pooled tokens vs FiLM/adaptive-LN). Add a tiny diagram with layer shapes.

- The authors need to clarify whether captions are generated online at inference or pre-computed offline. If online, please report end-to-end latency per pair, broken down into: (i) caption generation, (ii) text encoding, (iii) visual forward, (iv) geometric–semantic refinement. A table with mean ± std over N pairs and hardware specs (GPU/CPU, batch size) would help practitioners gauge deploy ability.

- To understand robustness, please swap GPT-4o for at least one open multimodal model (e.g., LLaVA-Next, InternVL, Qwen-VL) using the same prompts, and report the delta on your main metrics. An ablation that varies prompt temperature/length and a “noisy caption” stress test (paraphrases, truncated text) would quantify how brittle/forgiving the fusion is.

- In the SAM2 ↔ Grounded-SAM agreement, please list the exact IoU/overlap thresholds (αₜ) you used for positive matches and any NMS/size filters. Include a sweep (e.g., αₜ ∈ {0.3, 0.4, 0.5, 0.6}) with performance vs. retained proposals to show the precision–recall trade-off and your chosen operating point.

- Describe how you verify the common-view estimate quantitatively, inlier ratios from two-view matching, reprojection error, or success rates against human labels. Please discuss failure modes under extreme parallax, rolling shutter, or large time gaps, and provide a small error analysis with examples.

- Kindly include: (i) the NYC-CD dataset license, (ii) a brief privacy statement detailing face/license-plate handling and any additional redactions, and (iii) terms for using NYU-VPR imagery and GPT-4o-generated captions in annotations (ownership, redistribution, and commercial use). A one-page DATA & ETHICS note would make reuse straightforward.

---

### Official Review · Reviewer_aW89 · 2025-10-31

**Soundness:** 3
**Presentation:** 3
**Contribution:** 2
**Rating:** 2
**Confidence:** 5

**Summary:**

This paper proposes a vision-language framework for scene change detection by incorporating semantic understanding through language. For training of the proposed method, this papser also proposes semi-automatic annotation pipeline and builds a new dataset NYC-CD with multiclass change annotations.

**Strengths:**

(1) The paper is well-structured and easy to follow.

(2) This paper proposes semi-automatic annotation pipeline and builds a new dataset NYC-CD with multiclass change annotations.

**Weaknesses:**

(1) Application scenarios are limited, since the proposed method introduces Multimodal LLM/GPT-4o generates descriptions of changed objects in Image 1 relative to Image 0.

(2) The idea of this paper is similar to the following reference [1], only with different application fields.
	[1] Change Knowledge-Guided Vision-Language Remote Sensing Change Detection, IEEE TGRS, 2025.

(3) Lack of efficiency evaluation of methods. Due to the introduction of MLLM/GPT-4o, the efficiency of the proposed method should be relatively low and difficult to apply in real-world scenarios.

**Questions:**

To construct meaningful image pairs for change detection, the authors employ visual place recognition (VPR) instead of simple GPS matching. However, for real-world applications, should the NYC-CD dataset and proposed method be able to identify image pairs with meaningless changes? Since using GPS for positioning is often the most convenient and efficient method.

---

### Official Review · Reviewer_Hoop · 2025-11-02

**Soundness:** 3
**Presentation:** 3
**Contribution:** 2
**Rating:** 4
**Confidence:** 4

**Summary:**

This paper introduces a vision-language framework mainly for scene change detection (SCD) in urban environments, to address challenges such as lighting and viewpoint variations by fusing visual features with semantic language descriptions from Vision-Language Models (VLMs). The proposed architecture includes a feature enhancer with text features and a geometric-semantic matching module, along with a new NYC-CD dataset selected and annotated from the NYU-VPR dataset. Experimental results on three datasets demonstrate the effectiveness of the proposed modules.

**Strengths:**

* Introduces a novel dataset that introduce new target and non-target changes, created using a semi-automatic pipeline based on VLMs, segmentation and reconstruction foundation models.
* Proposes a scalable semi-automatic annotation pipeline that efficiently generates annotations for training.
* Experimental results show performance gains from the proposed method in most settings.

**Weaknesses:**

* Although a new architecture and dataset are proposed, the contributions and insights are not sufficiently clear. If the method is the primary contribution, it would be better to clarify its details and conduct more extensive experiments to demonstrate generalizability across additional datasets, as well as the impact of each module on performance.
* The motivation for the proposed dataset is unclear (despite mentions of constructing meaningful pairs to enable comprehensive evaluation, and descriptions in lines 300-311). What are its main advantages and how do the results support them? For example, motion blur is considered a non-target change, how does this affect the performance or evaluation of the methods? Additionally, what is the ratio of target to non-target changes in the new dataset?
* No ablation study is performed on the geometric and semantic matching modules.
* Since the proposed method incorporates foundation models, it would be beneficial to provide a runtime report, including GPU types, inference time, FPS and similar metrics.

**Questions:**

* Could authors provide qualitative examples of the descriptions and segmentations of initial results generated by the VLM/SAM2?
* Is the proposed method primarily two modules, or is there a standard method involved?
* How are the proposed methods combined with other approaches to obtain the results in Table 2?
* Is SAM2 indispensable for obtaining the initial semantic masks in the application of the proposed method, even during inference?
* How does the performance of SAM2 and GPT-4o affect the final SCD results? Is it possible that failures in rubble segmentation could lead to a decrease in performance on the PSCD dataset?

---

### Note · Authors · 2025-11-14

I have read and agree with the venue's withdrawal policy on behalf of myself and my co-authors.